# Production, Characterization, and Assessment of Permanently Cationic and Ionizable Lipid Nanoparticles for Use in the Delivery of Self-Amplifying RNA Vaccines

**DOI:** 10.3390/pharmaceutics15041173

**Published:** 2023-04-07

**Authors:** Dylan Kairuz, Nazia Samudh, Abdullah Ely, Patrick Arbuthnot, Kristie Bloom

**Affiliations:** Wits/SAMRC Antiviral Gene Therapy Research Unit, Infectious Diseases and Oncology Research Institute (IDORI), Faculty of Health Sciences, University of the Witwatersrand, Johannesburg 2050, South Africa2297183@students.wits.ac.za (N.S.); abdullah.ely@wits.ac.za (A.E.);

**Keywords:** saRNA, RNA delivery, cationic lipid nanoparticles, lipid film hydration, microfluidics, RNA vaccines, LNP production, LNP characterization

## Abstract

Africa bears the highest burden of infectious diseases, yet the continent is heavily reliant on First World countries for the development and supply of life-saving vaccines. The COVID-19 pandemic was a stark reminder of Africa’s vaccine dependence and since then great interest has been generated in establishing mRNA vaccine manufacturing capabilities on the African continent. Herein, we explore alphavirus-based self-amplifying RNAs (saRNAs) delivered by lipid nanoparticles (LNPs) as an alternative to the conventional mRNA vaccine platform. The approach is intended to produce dose-sparing vaccines which could assist resource-constrained countries to achieve vaccine independence. Protocols to synthesize high-quality saRNAs were optimized and in vitro expression of reporter proteins encoded by saRNAs was achieved at low doses and observed for an extended period. Permanently cationic or ionizable LNPs (cLNPs and iLNPs, respectively) were successfully produced, incorporating saRNAs either exteriorly (saRNA-Ext-LNPs) or interiorly (saRNA-Int-LNPs). DOTAP and DOTMA saRNA-Ext-cLNPs performed best and were generally below 200 nm with good PDIs (<0.3). DOTAP and DDA saRNA-Int-cLNPs performed optimally, allowing for saRNA amplification. These were slightly larger, with higher PDIs as a result of the method used, which will require further optimization. In both cases, the N:P ratio and lipid molar ratio had a distinct effect on saRNA expression kinetics, and RNA was encapsulated at high percentages of >90%. These LNPs allow the delivery of saRNA with no significant toxicity. The optimization of saRNA production and identification of potential LNP candidates will facilitate saRNA vaccine and therapeutic development. The dose-sparing properties, versatility, and manufacturing simplicity of the saRNA platform will facilitate a rapid response to future pandemics.

## 1. Introduction

Traditional vaccines have had a significant global impact on public health and the economy by reducing the spread of infectious diseases and their associated mortalities [1]. However, the lack of effective vaccines for some diseases, cell culture-dependent manufacturing, and an inability to respond rapidly to disease outbreaks and emerging variants has led to the development of next-generation nucleic acid-based vaccines, which are easily produced using cell-free techniques. In some instances, these vaccines are more efficient at eliciting protective immune responses [2]. However, Africa’s reliance on First World countries for vaccines has severely limited the continent’s ability to respond to disease outbreaks leaving it vulnerable to vaccine inequity. The World Health Organization, together with international and local collaborators, has taken the first steps to establish an mRNA technology transfer program which is intended to enable African countries to produce and supply mRNA-based vaccines to the continent. The benefits of this vaccine platform have been recently showcased with the rapid development of multiple, highly effective vaccines against SARS-CoV2 and numerous other pathogens [3]. mRNA vaccines enable in situ translation of antigens ensuring that antigens are folded in their native conformations and bear the correct post-translational modifications. These qualities help guide humoral and cellular immunity to ensure robust immune responses. 

The transient nature of antigen expression achieved by conventional mRNA vaccines means that large amounts of mRNA are required to ensure the production of adequate amounts of antigen for an effective immune response [4]. This represents a manufacturing bottleneck, especially for developing countries aiming to achieve vaccine independence. In this regard, the synthetic self-amplifying RNA (saRNA) vaccine platform provides an attractive alternative to conventional mRNA vaccines. This sophisticated approach exploits the self-replicating properties of alphaviruses, such as Sindbis virus (SINV), Semliki forest virus (SFV) and Venezuelan Equine Encephalitis virus (VEEV), to enable in situ replication of mRNA sequences that encode vaccine antigens [2]. saRNA vaccine constructs encode the four alphavirus-derived non-structural proteins (nsp1-4) which form the RNA-dependent RNA polymerase (RDRP) complex required for in situ self-propagation. The sequence encoding the vaccine antigen is included downstream, under the control of a subgenomic promoter, and is exponentially amplified. This ensures the expression of large amounts of antigen from smaller doses of saRNAs when compared to conventional mRNAs [4]. saRNAs, and the double-stranded replication intermediates that are formed in situ, also possess the inherent potential to trigger an innate immune response and are thus unlikely to require co-formulation with adjuvants for vaccine production [5]. 

Unformulated RNA vaccines are susceptible to RNase degradation, resulting in reduced vaccine efficacy [6]. To protect synthetic mRNA and facilitate cellular uptake, numerous non-viral nanoparticles, including lipid nanoparticles (LNPs), have been explored. To date, LNPs are the most commonly utilized platform for the delivery of mRNAs [7] and saRNAs [8]. Two subsets of LNPs have been defined: ionizable (iLNPs) and cationic LNPs (cLNPs). iLNPs consist of electron-dense core structures and are positively charged at a low pH [9], whereas cLNPs consist of lipid bilayers that are permanently cationic [10]. This positive charge facilitates the encapsulation or adsorption of negatively charged mRNA transcripts. Although iLNPs have been more extensively used for the delivery of saRNA vaccines, their high production cost limits their use in resource-constrained countries. On the other hand, cLNPs are cheaper and represent a promising alternative to other non-viral vectors; they have recently been explored as vectors for saRNA vaccines [11,12,13,14,15].

In this study, we describe the foundations for assessing LNP-formulations for the delivery of in vitro transcribed saRNA transcripts. Herein, we report on the synthesis of saRNAs encoding reporter proteins, the optimization of LNP-formulations for both the encapsulation and adsorption of saRNA, the kinetics of in vitro protein expression, and the effects of saRNAs on innate immunostimulation.

## 2. Materials and Methods

### 2.1. Synthesis of saRNAs Encoding Reporter Proteins

Sequences encoding reporter proteins eGFP and luciferase 2 (Luc 2) (Promega, Madison, WI, USA) were subcloned into a pUC57-based backbone upstream of a 3′ untranslated region, an extended poly-A tail and *Mlu*I restriction site. These plasmids were then digested with *Nde*I and *Mlu*I to excise the insert sequence encoding eGFP/Luc2, 3′ untranslated region, and poly-A tail. A T7-VEE-GFP-IRES-Puro plasmid was used for the cloning of the eGFP/Luc2 saRNA encoding plasmids [16]. This plasmid was a gift from Steven Dowdy (Addgene plasmid # 58977; http://n2t.net/addgene:58977 (last accessed on 4 April 2023); RRID: Addgene_58977). T7-VEE-GFP-IRES-puro plasmid was digested with *Nde*I and *Mlu*I to remove the GFP-IRES-puro sequences. Backbone and insert sequences were ligated at a 1:3 ratio to produce the single-cistronic saRNA plasmids T7-VEEV-eGFP and T7-VEEV-Luc2. Plasmids were propagated overnight in NEB Turbo competent *E. coli* (New England Biolabs, Ipswich, MA, USA). This was followed by plasmid extraction and purification using a Qiagen^®^ plasmid Maxi Kit (Qiagen, Hilden, Germany), according to the manufacturer’s instructions. Plasmids were verified by restriction digest and sanger sequencing (Inqaba Biotec, Pretoria, South Africa).

The T7-VEEV-eGFP and T7-VEEV-Luc2 plasmids were linearized immediately downstream of the poly-A tail using the restriction enzyme *Mlu*I and used as templates for the in vitro transcription (IVT) of saRNA-eGFP and saRNA-Luc2, respectively. SaRNAs were transcribed using a TranscriptAid T7 High Yield transcription kit (Thermo Fisher Scientific, Waltham, MA, USA). The capping of transcripts (cap 1) was performed using a Vaccinia capping system (New England Biolabs, Ipswich, MA, USA) and mRNA cap 2′-O-methyltransferase (New England Biolabs, Ipswich, MA, USA), according to the manufacturer’s instructions. In addition, the IVT of saRNA-eGFP and saRNA-Luc2 was performed using a transcription buffer optimized for the synthesis of longer RNAs, as previously described [17]. Each 100 µL IVT reaction included 10 mM NTPs (New England Biolabs, Ipswich, MA, USA), 100 ng/µL linearized template, 0.04 U/µL Murine RNase inhibitor (New England Biolabs, Ipswich, MA, USA), and 8 U/µL T7 RNA polymerase (New England Biolabs, Ipswich, MA, USA) in 1× IVT buffer. These transcripts were co-transcriptionally capped using either Anti-Reverse Cap analogue (ARCA) (New England Biolabs, Ipswich, MA, USA) or CleanCap^®^ AU reagent (TriLink Bio Technologies, San Diego, CA, USA) at a final concentration of 10 mM or 8 mM, respectively. Reactions were incubated at 37 °C for 2 h, followed by DNase I treatment (1 U/µg DNA template in 1× DNase buffer) for 20 min at 37 °C. Transcripts were precipitated by the addition of lithium chloride to a final concentration of 2.5 M and incubated at −20 °C for 30 min. Transcripts were collected by centrifugation (12,000 rpm for 15 min at 4 °C). Pellets were washed in 75% ethanol, centrifuged (12,000 rpm for 10 min at 4 °C), and resuspended in nuclease-free water. RNA was aliquoted and stored at −80 °C.

The concentration of saRNAs was determined by spectrophotometry using a NanoPhotometer^®^ (Implen, Westlake Village, CA, USA). The size and integrity of transcripts were determined by denaturing formaldehyde gel electrophoresis. Briefly, 1 µg of saRNA in 1× RNA loading dye (New England Biolabs, Ipswich, MA, USA) was heated at 70 °C for 10 min before loading onto a 1% denaturing agarose gel. A 200 to 6000 base RiboRuler High Range RNA Ladder (Thermo Fisher Scientific, Waltham, MA, USA) was included as an RNA size marker. Denaturing agarose gels were run at 80 volts for 45 min before visualization on an Omega Fluor Gel Documentation System (Aplegen, Pleasanton, CA, USA). 

### 2.2. Transfection of HEK 293T Cells with saRNAs and Detection of Reporter Proteins

HEK 293T cells were grown in complete media (high glucose (4.5 g/L) Dulbecco’s Modified Eagle Medium supplemented with 10% foetal calf serum, 100 U/mL penicillin, and 100 µg/mL streptomycin) and maintained in a humidified incubator at 37 °C and 5% CO_2_. Prior to transfection, cells were seeded at a confluency of 40% in 48-well plates. Cells were transfected with 200 ng of saRNAs using commercially available liposome-based Lipofectamine™ MessengerMAX™ transfection reagent (Invitrogen, Waltham, MA, USA) at an RNA (µg): Lipofectamine™ (µL) ratio of 1:1.5, or LNP formulations, as described in Section 3.3, Section 3.4 and Section 3.5. The expression of fluorescent reporter proteins was examined at various timepoints using an EVOS Fluorescence Microscope (Invitrogen, Waltham, MA, USA). The expression of Luc2 was examined using a Luciferase^®^ Reporter Assay System kit (Promega, Madison, WI, USA), according to the manufacturer’s instructions. Briefly, at 24 or 48 h post-transfection, cells were lysed by incubation in passive lysis buffer with shaking for 15 min at room temperature. LARII substrate was added to cell lysates and relative light units were detected using GloMax^®^ Explorer (Promega, Madison, WI, USA). 

### 2.3. Quantitative Reverse Transcription-Polymerase Chain Reaction (qRT-PCR)

HEK 293T cells were seeded at 40% confluency in 6-well plates and transfected with 250, 500 or 1000 ng of saRNA-Luc2 or Poly I:C (double-stranded RNA positive control) using Lipofectamine™ MessengerMAX™, as previously described. Cells transfected with Lipofectamine™ MessengerMAX™ alone (mock transfections) served as a negative control. At 24 h post transfection, the supernatant was removed, and the total RNA was extracted from cells using TRIzol™ reagent (Invitrogen, Waltham, MA, USA), according to the manufacturer’s instructions. Precipitated RNA was resuspended in 85 µL of nuclease-free water and treated with 5 units of DNaseI (Thermo Fisher Scientific, Waltham, MA, USA) in 1× DNaseI buffer at 37 °C for an hour to remove contaminating DNA. RNA was precipitated again using 75% ethanol and resuspended in nuclease-free water. The concentration and purity of the total RNA were analysed using a NanoPhotometer^®^ (Implen, Westlake Village, CA, USA). The presence of DNA contaminants and the integrity of RNA were assessed by agarose gel electrophoresis, as previously described. RNA was stored at –80 °C.

A LUNA^®^ Universal One-Step RT-qPCR kit (New England Biolabs, Ipswich, MA, USA) was used to reverse transcribe the total RNA, after which cDNA encoding *Interferon-β* (*INF-β*), *2′-5′-oligoadenylate synthetase 1* (*OAS1*), *interferon-induced proteins with tetratricopeptide repeats 1* (*IFIT1*), *Protein Kinase R* (*PKR*) and the housekeeping gene *glyceraldehyde 3-phosphate dehydrogenase* (*GAPDH*) were amplified using gene-specific primers (Appendix A). Each 20 µL reaction mix contained 500 ng total RNA, 1× Luna WarmStart RT enzyme mix, 1× Luna Universal One-Step Reaction mix, 0.4 µM of each forward and reverse primer, and nuclease-free water. Thermocycling was performed using a Biorad C1000 Touch thermocycler with a CFX96 real-time PCR detection system (Bio-Rad Laboratories, Hercules, CA, USA). Thermocycling conditions consisted of reverse transcription at 55 °C for 10 min and initial denaturation at 95 °C for 60 s. This was followed by 40 cycles of denaturation at 95 °C for 10 s, annealing at 58 °C for 30 s and extension at 60 °C for 60 s. The specificity of amplification was verified by melt-curve analysis (60–95 °C) and agarose gel electrophoresis. mRNA expression levels were normalized to *GAPDH* mRNA, and the relative fold change in expression between transfected and mock-transfected samples was determined using the delta delta Ct method [18].

### 2.4. Lipid Nanoparticle Formulation of saRNAs

DOTAP (1,2-dioleoyl-3-trimethylammonium-propane), DOTMA (1,2-di-O-octadecenyl-3-trimethylammonium propane), DDA (Dimethyldioctadecylammonium), DSPC (1,2-distearoyl-sn-glycero-3-phosphocholine), DOPE (1,2-dioleoyl-sn-glycero-3-phosphoethanolamine) and DMG-PEG 2000 (1,2-dimyristoyl-rac-glycero-3-methoxypolyethylene glycol-2000) were obtained from Avanti Polar Lipids, Inc. (Alabaster, AL, USA). Cholesterol was obtained from Sigma-Aldrich (St. Louis, MI, USA). RNA-LNPs were formulated using either lipid film hydration (LFH) or a modified solvent injection method. Multi-component LNP lipid formulation stocks (20 mg/mL) were pre-mixed and stored at −20 °C (Appendix A). These were diluted to 4–10 mg/mL in ethanol to a final volume of 0.5–1 mL. Lipid films were produced by ethanol evaporation using nitrogen under spiral flow with a Smart Evaporator C1 (BioChromato, Inc., Kanagawa, Japan). This was followed by vacuum desiccation for 2 h to remove residual ethanol. Lipid films were then hydrated with pre-warmed HEPES buffered saline (HBS) for 30 min, followed by overnight incubation at 4 °C to ensure sufficient hydration. To create unilamellar vesicles and reduce LNP size, bath sonication and extrusion were performed. Extrusion was accomplished using an Avanti^®^ Mini Extruder (Avanti Polar Lipids, Inc., Alabaster, AL, USA), according to the manufacturer’s instructions. LNPs were equilibrated to a temperature above their phase transition temperature and extruded a minimum of 11× to ensure a uniform size distribution. This was performed sequentially through polycarbonate membranes from higher to lower pore sizes ranging from 400 nm–100 nm. saRNAs were complexed externally by mixing with LNPs at different N:P ratios (positive nitrogen on the cationic lipid:negative phosphate on the RNA) and incubating the mixture at room temperature for 30–45 min before use. 

Precision NanoSystems Ignite with NxGen Cartridges (Precision NanoSystems Inc., Vancouver, BC, Canada) was used to formulate iLNPs using microfluidics. saRNAs diluted in an acidic buffer (100 mM citrate buffer, pH 4) and lipid mixes dissolved in ethanol were mixed at a total flow rate of 5 mL/min, and a flow rate ratio of 3:1. Dlin-DMA-MC3 (DC Chemicals, Shanghai, China), Cholesterol (Sigma-Aldrich, St. Louis, MI, USA), DSPC, and DMG-PEG2000 (Avanti Polar Lipids, Inc., Alabaster, AL, USA) were formulated at a molar ratio of 50:38.5:10:1.5 and an N:P ratio of 8:1.

The modified solvent injection [11,19] method was performed by diluting saRNA in an aqueous buffer (100 mM Tris pH 7). Lipids dissolved in ethanol were then injected into the diluted saRNAs. The mix was then vortexed to rapidly mix the components and encapsulate the saRNAs.

The resulting LNPs were either dialyzed to remove excess ethanol, or diluted 40× in the buffer of interest (e.g., PBS) and concentrated using Amicon^®^ Ultra 15 (Millipore Sigma, Burlington, MA, USA) concentrator columns. Dialysis was performed for 2 h against the buffer of interest. Particle sizes were measured using the ZetaSizer Pro Blue (Malvern Panalytical, Malvern, UK). saRNA-LNPs were stored at 4 °C. 

For transfections, 200 ng and 100 ng of saRNA-LNPs diluted in OptiMEM (Thermo Fisher Scientific, Waltham, MA, USA) were used to transfect HEK293 cells seeded at 40% confluency in 48-well and 96-well plates, respectively. HEK293 cells were maintained and transfected in the presence of complete media (as described in Section 2.2). 

### 2.5. saRNA Quantification and Encapsulation Efficiency Assessment

To quantify saRNAs complexed to LNPs (effective dose) and assess encapsulation efficiency in interiorly and exteriorly formulated saRNA-LNPs, a Quant-iT RiboGreen RNA Kit (Thermo Fisher Scientific, Waltham, MA, USA) was used. Samples were diluted either in TE buffer (pH 7.5) or in Triton X-100 buffer and incubated at 37 °C for 10 min to lyse LNPs in the presence of the Triton X-100 detergent. This allows quantification of uncomplexed RNA (TE Buffer) and total RNA (complexed and uncomplexed, Triton X-100 buffer), and hence the percentage of RNA stably complexed to the LNPs can be determined. Ribogreen reagent was then added to each well and fluorescence was measured using GloMax^®^ Explorer (Promega, Madison, WI, USA). A standard curve was generated (0.1–2.5 ng/µL final RNA concentration) in the presence of Triton X-100. Encapsulation efficiencies and effective saRNA concentrations were calculated using the following equations, where total saRNA (µg/mL) was calculated using the standard curve:
$$\mathrm{Encapsulation}\;\mathrm{Efficiency}\;(\%)=\;\left(1-\frac{\mathrm{Fluorescence}\;\mathrm{Value}\;\mathrm{of}\;\mathrm{saRNA}\;\mathrm{LNPs}\;\mathrm{Lysed}\;\mathrm{with}\;\mathrm{Triton}\;\mathrm{Buffer}}{\mathrm{Fluorescence}\;\mathrm{Value}\;\mathrm{of}\;\mathrm{saRNA}\;\mathrm{LNPs}\;\mathrm{in}\;\mathrm{the}\;\mathrm{Absence}\;\mathrm{of}\;\mathrm{Lysis}}\right)\times 100$$
Effective Concentration (μg/mL)=Total saRNA Concentration (μg/mL)×Encapsulation Efficiency (%)

### 2.6. Cell Viability Assay

Potential toxicity of saRNAs and LNPs was assessed using an MTT [3-(4, 5-dimethylthiazol-2-yl)-2, 5-diphenyltetrazolium bromide] assay (Sigma-Aldrich, St. Louis, MI, USA). HEK 293T cells were seeded in 96-well plates and transfected with 125 ng of saRNAs formulated with LNPs or Lipofectamine™ MessengerMAX™ transfection reagent. Cells transfected with delivery vehicle alone (empty cLNP/Lipofectamine) served as controls, and DMSO treatment for 30 min was included as a control for cellular toxicity. MTT assays were performed 24 h post transfection, and absorbance was measured at 570 nm using an iMark microplate reader (BioRad, Hercules, CA, USA).

### 2.7. Statistical Analysis

Results are displayed as the mean ± standard error of the mean. Graphs were generated using GraphPad Prism 4/5, or Microsoft Excel. Statistical differences were analyzed using a Student’s *t*-test with *p* < 0.05 indicating significance.

## 3. Results

### 3.1. IVT of saRNAs Encoding Reporter Proteins

SaRNAs encoding reporter proteins eGFP or Luc2 were synthesized by IVT (Figure 1a). Although good yields of saRNAs were obtained using the TranscriptAid T7 High Yield kit, size and integrity analysis by formaldehyde gel electrophoresis revealed the presence of incomplete or degraded transcripts (Figure 1b). Post-transcriptional capping using the Vaccinia capping system to produce saRNA transcripts with cap 1 structures, for improved in vivo translation, resulted in further degradation of the saRNA transcripts. Comparatively, conventional mRNA encoding eGFP appeared intact even following post-transcriptional capping (Figure 1b). Commercially available IVT kits are optimized for the synthesis of conventional mRNAs. However, saRNA transcripts are significantly longer than conventional mRNAs because they encode the antigen of interest as well as an approximately 7.5-kilobase sequence encoding VEEV nsPs 1-4. Increased handling of such long transcripts by post-transcriptional capping methods increases the likelihood of RNA degradation. It is also inevitable that during IVT, some incomplete transcripts will be present at the end of the incubation period. For these reasons, an IVT buffer optimized for the synthesis of longer saRNAs, and containing a higher concentration of magnesium acetate, was prepared. This type of buffer, with an NTP:magnesium ion ratio of 1:1.875, has been reported to improve the yield of saRNA transcripts [17]. To further increase saRNA yield, the concentration of T7 polymerase was increased to 8 units/µL. To reduce handling and degradation of saRNAs, co-transcriptional capping, using ARCA (cap 0) or CleanCap^®^ AU reagent (cap 1), was performed instead of post-transcriptional capping. These modifications resulted in a higher proportion of full-length saRNA transcripts (Figure 1c). 

Fluorescence microscopy was used to confirm expression of eGFP from ARCA and CleanCap^®^ AU saRNA-eGFP transcripts at 24 h (day one), day two and day five after transfection of HEK 293T cells using commercially available Lipofectamine™ MessengerMAX™. Strong expression of eGFP from both transcripts was still observed 5 days after transfection (Figure 1d). Expression of Luc2 from both ARCA and CleanCap^®^ saRNA-Luc2 transcripts was also observed in HEK 293T cells with CleanCap^®^ transcripts clearly showing a significantly greater expression compared to ARCA transcripts, and comparable to expression from an equivalent amount of Luc2-encoding plasmid DNA (Figure 1e).

### 3.2. Innate Immune Response to saRNAs

In situ self-replication of saRNAs results in the production of double-stranded RNA replication intermediates which trigger the innate immune system, resulting in the secretion of cytokines conducive to the development of a Th1 immune response [20]. Although this may enhance immunogenicity of the vaccine, an innate immune system response may also prematurely inhibit translation of the antigen to diminish vaccine efficacy. This occurs when interferon-induced proteins create an antiviral state whereby protein translation is halted, RNA is degraded, and apoptosis is initiated [5]. 

Dose response assays to determine the extent to which saRNAs induce a type 1 interferon response were performed. Induction of an interferon response was examined by qRT-PCR to measure the fold change in the mRNA concentrations of *INF-β* and interferon-inducible genes in transfected cells relative to mock-transfected control cell cultures (Figure 2; Appendix A). As expected, the highest dose (1000 ng) of saRNA-Luc2 induced an increase in mRNA transcripts encoding *IFN-β* (5.5-fold), *IFIT1* (2.5-fold) and *OAS1* (13.8-fold). No increase in *IFN-β* was observed at the 500 ng and 250 ng doses; however, an increase in *OAS1* transcripts (5.83-fold and 1.88-fold, respectively) was observed, suggesting that IFN-β had been expressed. The positive control, Poly I:C, induced a much stronger interferon response. At the highest dose, Poly I:C induced significant increases in transcripts encoding *OAS1* (647.6-fold), *IFN-β* (51.44-fold), and *IFIT1* (31.46-fold). The 500 ng dose elicited significant increases in transcripts encoding *OAS1* (208.63-fold) and *IFIT1* (3.41-fold). This is expected because saRNAs are much longer compared to Poly I:C and therefore an equivalent concentration of saRNAs would contain fewer molecules to activate pattern recognition receptors. mRNAs encoding PKR were not significantly upregulated at any of the doses tested for all constructs including the Poly I:C positive control. Thus, we can conclude that these saRNAs stimulate the innate immune system in a dose-dependent manner; however, this does not appear to hamper the expression of reporter proteins in vitro because large amounts of eGFP and Luc2 were observed (Figure 1d,e).

### 3.3. External Formulation of saRNA-cLNPs Using Lipid Film Hydration

Two main types of LNPs are used to deliver RNAs: permanently cationic LNPs, where RNAs can be complexed internally (Figure 3a, top, saRNA-Ext-cLNPs) or adsorbed externally (Figure 3a, middle, saRNA-cLNP-Int), and ionizable LNPs (Figure 3a, bottom). The focus of this research was cLNPs. Initially, empty cLNPs with diameters less than 150 nm and polydispersity indices (PDIs) below the cut-off of 0.3 were produced by LFH (Figure 3b). Formulations were coded to help distinguish different LNPs according to the type of phospholipid used and the ratio of the lipids (i.e., F1 and F2 are the same ratio using DOPE and DSPC, respectively; Appendix A). CaliVax, a commercially available cLNP (~100 nm, DOTAP:Cholesterol 1:1) was used for comparison. As expected, the addition of saRNAs (external formulation by adsorption of saRNAs) increased the LNP size with a decreasing N:P ratio (Figure 3c). LNPs comprising different lipids in different ratios were produced with the main cationic lipids being DOTAP, DDA and DOTMA. Sizes of saRNA-Ext-cLNPs were generally below 200 nm (Figure 3d), with good PDIs (Figure 3e); however, some displayed larger sizes with PDIs at or above the cut-off of 0.3 (Appendix A). 

### 3.4. Optimization of Externally Formulated saRNA-cLNPs for In Vitro Expression

Different saRNA-Ext-cLNPs were formulated at different N:P ratios to optimize saRNA-eGFP expression and determine potential delivery candidates for future potential vaccines. In situ translation of eGFP in HEK293 cells following transfection of different saRNA-Ext-cLNP formulations showed that the N:P ratio (Figure 4a) and cLNP composition (Figure 4b) were more important determinants of delivery efficacy than size. DOTMA F1 cLNPs showed improved efficacy at a higher N:P (12:1) ratio, whereas CaliVax performed optimally at a low (2.5:1) ratio (Appendix A), despite being much larger (347 nm). This expression was maintained from 24–48 h, which is expected as a result of saRNA replication over time. A library of saRNA-Ext-cLNPs was optimized based on N:P ratios, showing the effect of different LNP lipid composition and molar percentage on RNA delivery (Figure 4b). DOTMA saRNA-Ext-cLNPs appear to be the most efficient at RNA delivery, followed by DOTAP, especially the “F1” and “F2” formulations. This was followed by the “G1” formulations, indicating that an increased molar percentage of the main cationic (complexing) lipid appears to be a significant factor in improved saRNA delivery. Although DDA saRNA-Ext-cLNPs did not perform as well as DOTAP and DOTMA, previous studies have shown they are efficient vaccine delivery vehicles and may still be good candidates to explore further [12].

DOTAP F1, DOTMA F1 and CaliVax cLNPs displayed no obvious toxicity in cell culture, as measured by MTT assay (cell viability >75%, Figure 5a). Any cellular toxicity appeared to be a result of the combination of saRNAs complexed to cLNPs as neither saRNAs nor empty cLNPs alone resulted in any toxicity. Although % viability was reduced upon exposure to the combination of saRNAs and cLNPs as opposed to cLNPs alone, viability did not drop below 75%, and was significantly higher than the DMSO toxicity control (<5% viability). The encapsulation efficiency of these saRNA-cLNPs, which measures the percentage of saRNAs complexed with the LNPs, was optimal with >90% of saRNAs complexed (Figure 5b). DOTAP and DOTMA B1 also showed high levels of RNA complexation when examined using a gel retardation assay (Appendix A). This is most likely a result of the permanently positive charge of the main complexing lipid.

### 3.5. Internal Formulation of saRNA-cLNPs by a Modified Solvent Injection Method

Internal formulations, where saRNAs are encapsulated within the LNP (saRNA-Int-LNPs), were also investigated. Formulation was performed using a modified solvent injection method based on previous literature [11,19]. This method produced saRNA-cLNPs of varying sizes and size distributions, with PDIs generally above the 0.3 cut-off (Figure 6a, Appendix A). However, this is a crude method of production when compared to microfluidics, where precise flow rates can be specified. Further optimization of this method (formulation buffer composition, vortex time and speed, initial and final lipid concentrations) are likely to improve cLNP characteristics. Despite sub-optimal size and PDI, these saRNA-Fluc-cLNPs were able to effectively deliver the reporter transcript (Figure 6b). In particular, DOTAP B1 displayed an optimal PDI (<0.3), good size (<200 nm) and efficient delivery.

saRNA-Fluc2 expression kinetics differed between cLNP formulations, once again highlighting the importance of optimizing N:P ratios and lipid composition (Figure 6c). Although some saRNA-cLNPs, such as DOTAP F1 8:1, expressed well at 24 h, their expression had not increased significantly at 48 h suggesting that amplification of the transcript may have been impeded. Other formulations, such as DDA F2 12:1 and DOTAP F1 12:1, showed a 4- and 2.7-fold increase in expression from 24 to 48 h, respectively, which correlates with the expected expression kinetics of saRNAs. Surprisingly, an established ionizable LNP control (Dlin-DMA-MC3, 8:1 N:P) formulated using microfluidics did not show very efficient saRNA delivery, or amplification of expression from 24 to 48 h. This could be as a result of the iLNPs originally being optimized for encapsulation of significantly smaller siRNAs, as this LNP is used in Onpattro^®^ (Patisiran) [21,22]. Ribogreen assay analysis also showed a very high encapsulation efficiency of >90% using this technique (Figure 6d).

## 4. Discussion

The ability of mRNA vaccines to confer a protective immune response against SARS-CoV-2 has reignited the field of vaccinology and has brought the mRNA platform to the forefront of infectious disease vaccine development. This has been facilitated by the multiple iterations and recent advancements that synthetic RNA has undergone resulting in improved stability and translatability of mRNA vaccines and therapies [3]. A key advancement has been the use of modified nucleotides to prevent the recognition of synthetic exogenous mRNAs and induction of an innate immune response, leading to improved antigen expression. Unfortunately, these modifications cannot be applied to the saRNA vaccine platform, which relies on the formation of a 5′ stem-loop secondary structure, for in situ self-replication. Alterations to the thermodynamic stability of this secondary structure by uridine depletion or using modified nucleotides could have a negative impact on self-replication. These characteristics make saRNAs inherently immunogenic, which has raised concerns about whether enough antigen will be produced to achieve antigen-specific immunity and confer protection. Fortunately, the very same secondary structures enable saRNAs to replicate while evading the innate immune system [23], suggesting that the inherent ability of alphaviruses to control host responses could be favourable for vaccine development. 

Our data demonstrates that saRNAs encoding a reporter protein do indeed trigger the innate immune system in a dose-dependent manner, resulting in transcription of IFN-β, IFIT1 and especially OAS1, which activates RNase L, the enzyme responsible for degradation of cellular RNAs. Despite this, large amounts of reporter proteins were expressed in vitro, possibly because of the rapid self-replication and translation that occurs during the time it takes for interferon and interferon-stimulated genes to be transcribed and translated into effector proteins. Inhibition of translation of saRNAs by IFIT1 is not a concern because saRNAs possess a 5′ stem-loop structure that effectively blocks recognition and binding by IFIT1 [23]. It is more likely that after 24 h, saRNAs could be subject to degradation by RNase L. However, this degradation would be counteracted by the self-replication of saRNAs, ensuring that transcripts are available for translation. Despite the absence of modifications used to prolong the expression of proteins encoded in conventional mRNAs, strong expression of reporter proteins from saRNAs was observed for up to 5 days. For vaccine applications, this expression profile and innate immune system response would ensure the production of sufficient quantities of antigen while eliciting an effective immune response. 

Delivery of large saRNA transcripts remains an important consideration for the development of vaccines and therapies. In this study, multiple cLNPs were investigated for the delivery of saRNAs, yielding impressive cell culture expression results for both eGFP and Luc2 transcripts, accompanied by high encapsulation efficiency percentages for both external and internal formulations. LFH with bath sonication and extrusion was successfully used to produce empty cLNPs to which saRNA could be exteriorly complexed, and although it is not easily scalable, this method is a useful technique for LNP screening. We have developed multiple candidates which, through further investigations in vivo, will allow for the delineation of key saRNA-cLNP formulations to ensure cell culture data correlates with in vivo expression [19] and that antigenic protein expression results in vaccine efficacy [24]. SaRNA-Ext-cLNPs DOTAP and DOTMA F1 and F2 show the best expression in vitro which were accompanied by amplification of the bioluminescent signal. Although DDA showed lower expression, previously, it has been used for successful saRNA vaccine delivery and should still be examined in vivo [12,14]. A modified solvent injection method was also successfully applied to formulate saRNAs internally in cLNPs. DOTAP F1 and DDA F2 saRNA-Int-cLNPs showed the highest potential for in vivo use, mimicking the expected saRNA expression kinetics. These were able to outperform MC3-saRNA-iLNPs, which have previously been used to deliver saRNA vaccines with success [25]. Both methods of formulation yielded high saRNA encapsulation efficiencies and demonstrated the importance of optimizing factors such as N:P ratio and LNP composition. However, different LNP compositions may be more efficient at targeting different cell types [26]. This, along with variations in cell type immune response stimulation by different lipids [27], shows the importance of also examining alternative injection routes, as transfection and stimulation of certain cell populations may be important for different vaccines [26]. saRNA-LNPs were also shown to be safe in vitro, as no significant toxicity was observed. DOTAP has also been studied for use in RNA vaccines further supporting the safety and applicability of these results [12,14,15].

cLNPs have demonstrated strong potential for saRNA vaccine delivery and this has been recently explored with successes in preclinical saRNA vaccine studies, albeit less so than the more popular iLNPs. The lower cost of these cationic lipids is also an advantage over ionizable lipids, which can be expensive to synthesize. External formulation of saRNAs was achieved successfully, with particles forming the correct size and PDI parameters. Larger PDIs were present when using the modified solvent injection to formulate saRNA-Int-cLNPs; however, the optimization of buffer composition, vortex time and speed, initial and final lipid concentrations could improve this, which will be important for lower to middle-income countries, where access to microfluidics may be limited or impractical.

## Figures and Tables

**Figure 1 pharmaceutics-15-01173-f001:**
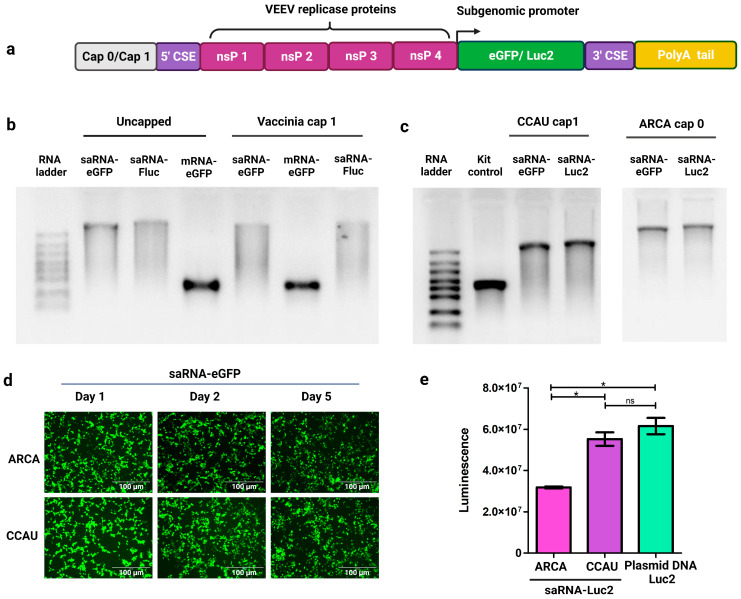
Design, synthesis, and expression of single-cistron saRNAs. (**a**) Schematic of capped saRNA transcript containing Venezuelan Equine Encephalitis Virus (VEEV)-derived sequences encoding non structural proteins 1-4 (nsP1-4) and conserved sequence elements (CSEs), reporter protein sequences under the control of the VEEV sub-genomic promoter, and poly-A tail. (**b**) Integrity and size analysis of saRNAs synthesized using the TranscriptAid T7 High Yield kit before and after capping with the Vaccinia capping system. (**c**) Integrity and size analysis of saRNAs synthesized using an optimized in vitro transcription buffer, and co-transcriptionally capped using ARCA or CleanCap^®^ AU (CCAU) reagent. The TranscriptAid T7 High Yield kit control was included as a conventional mRNA control. (**d**) Expression of eGFP in HEK 293T cells transfected with 200 ng of ARCA- or CCAU-saRNA-eGFP at days 1, 2 and 5 post-transfection. (**e**) Expression of luciferase in HEK 293T cells transfected with 200 ng of ARCA- or CCAU-saRNA-Luc2, or 200 ng of plasmid DNA encoding Luc2. Results are depicted as the mean, with standard error of the mean. Significant differences were calculated using a Student’s *t*-test. * *p* < 0.05, ns = not significant. Created using GraphPad Prism and BioRender.com.

**Figure 2 pharmaceutics-15-01173-f002:**
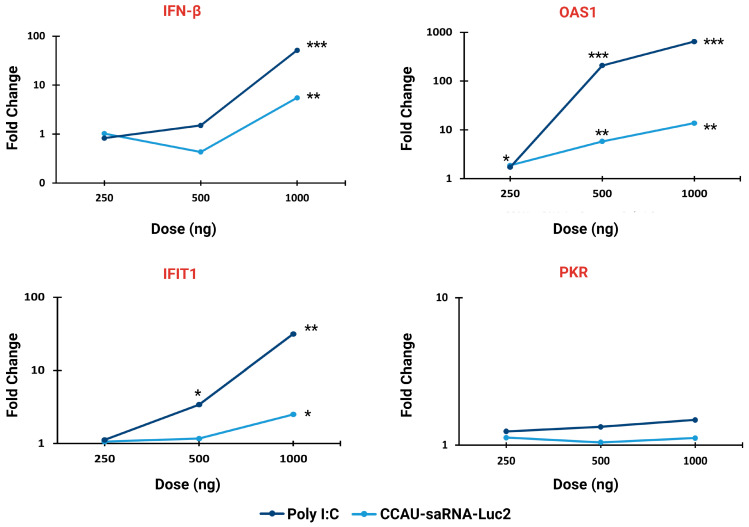
Innate immune response to saRNAs. Fold changes in mRNA transcripts encoding IFN-β and interferon-inducible genes (*OAS1*, *IFIT1*, *PKR*) in HEK 293T cells transfected with saRNA-Luc2, or Poly I:C. mRNA concentrations were determined using qRT-PCR and normalized to *glyceraldehyde 3-phosphate dehydrogenase* (*GAPDH*) expression. Relative fold change in expression between transfected and mock-transfected samples (n = 3) was determined using the delta delta Ct method. Poly I:C = positive control, INF-β = Interferon beta, OAS1 = 2′-5′-oligoadenylate synthetase 1, IFIT1 = interferon-induced proteins with tetratricopeptide repeats 1, PKR = Protein Kinase R. Significant differences between transfected and mock-transfected samples were calculated using a Student’s *t*-test. * *p* < 0.05, ** *p* < 0.01, *** *p* < 0.001, Created using Microsoft Excel and BioRender.com.

**Figure 3 pharmaceutics-15-01173-f003:**
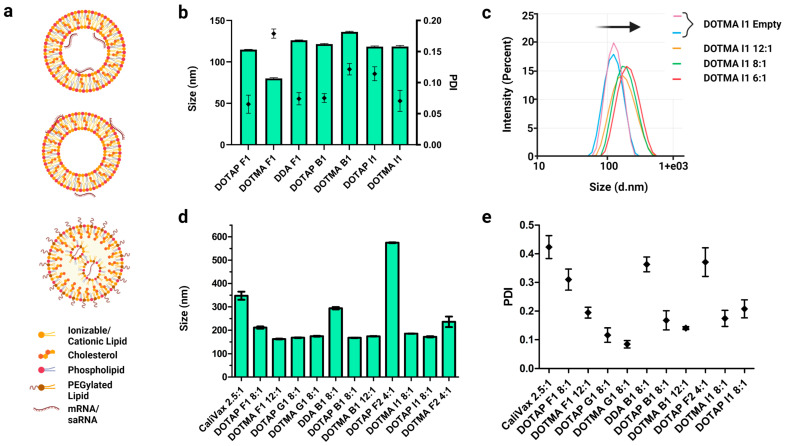
Production of cLNPs by lipid film hydration and external saRNA formulation. (**a**) cLNPs (top and middle) consist of a lipid bilayer, and RNAs can be complexed interiorly (top) or exteriorly (middle). These can also contain a PEGylated lipid. iLNPs (bottom) consist of an electron-dense structure. (**b**) Size and PDI were measured after bath sonication and extrusion. Although sizes were variable, all LNPs were below 150 nm with good PDIs (all < 0.2). (**c**) As saRNAs were added externally to LNPs, size increased with a decreasing N:P ratio (12:1 to 6:1) as shown by the size shift for the DOTMA I1 formulation (arrow). (**d**) When saRNAs were complexed externally, LNP sizes were generally ≤200 nm; however, outliers were present, especially at lower N:P ratios. (**e**) PDIs of externally formulated saRNA-cLNPs were generally ≤0.3. Lower N:P ratios generally produced higher PDIs. Results are depicted as the mean, with standard error of the mean. Created using GraphPad prism 4 and 5 and BioRender.com.

**Figure 4 pharmaceutics-15-01173-f004:**
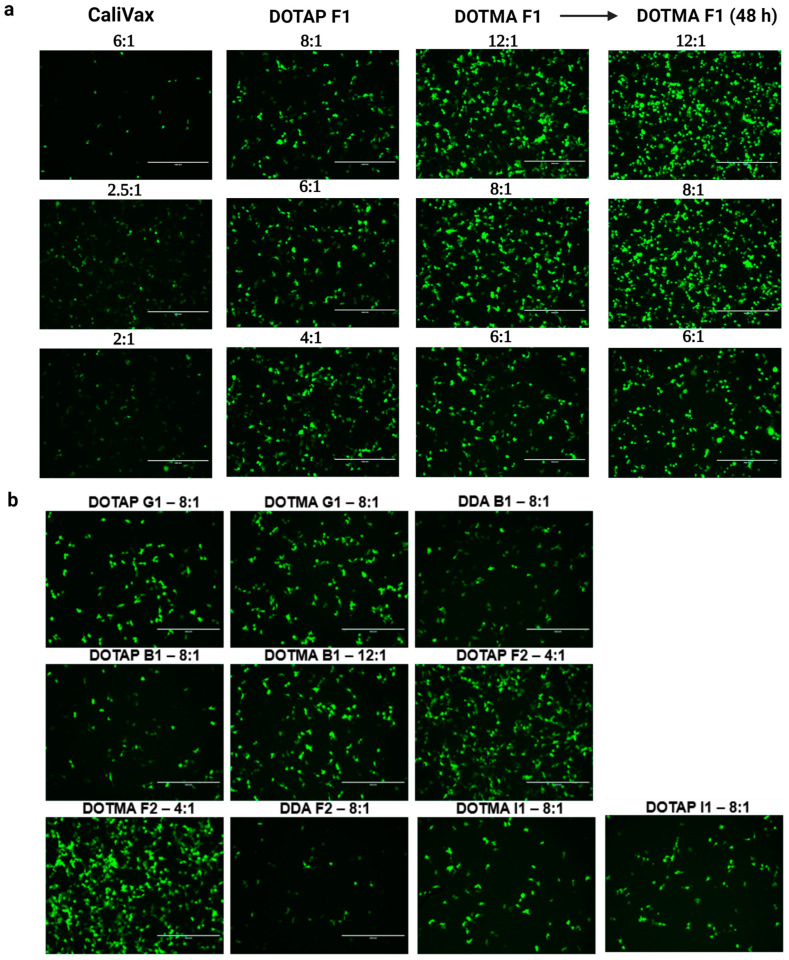
Optimization of saRNA-Ext-cLNPs produced using lipid film hydration. (**a**) Expression of eGFP following transfection of different saRNA-eGFP-cLNP formulations in HEK293 cells showing the effects of N:P ratio on different cLNPs. CaliVax nanoparticles performed best at lower ratios despite their larger size and PDI, whereas DOTAP and DOTMA F1 cLNPs performed optimally at higher N:P ratios. (**b**) A library of LNPs comprising DOTAP, DOTMA and DDA was optimized for delivery of saRNA-eGFP by changing the N:P ratios, showing the importance of optimizing N:P ratio, lipid composition and molar percentages. Scale bars on microscopy images represent 400 µm. Created using BioRender.com.

**Figure 5 pharmaceutics-15-01173-f005:**
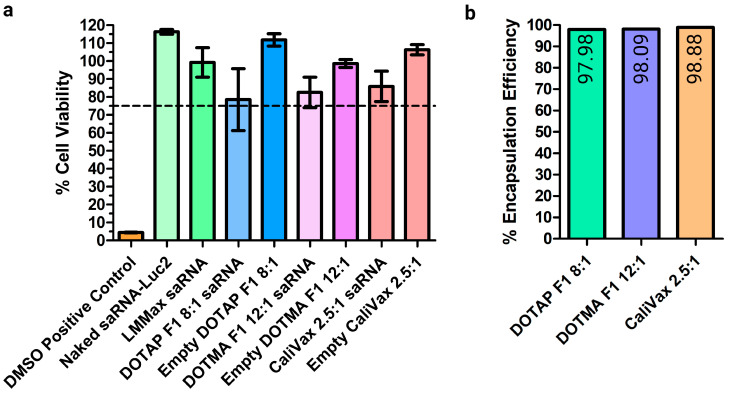
Absence of toxicity and high encapsulation efficiency of saRNA-Ext-cLNPs. (**a**) MTT cell toxicity assays were performed following transfection of saRNAs alone, cLNPs alone (empty) and formulated/complexed saRNA-cLNPs. Transfection of saRNAs with Lipofectamine Messenger Max (LMMax) and DMSO were included as controls. saRNA-cLNPs were found not to be toxic as percentage viability remained above 75%. (**b**) Encapsulation efficiency of CaliVax, and DOTMA and DOTAP F1 saRNA-cLNPs was >95% as measured by RiboGreen assay, showing strong adsorption of saRNA to the surface of the cLNPs. Results are depicted as the mean, with standard error of the mean. Created using GraphPad prism 4 and 5 and BioRender.com.

**Figure 6 pharmaceutics-15-01173-f006:**
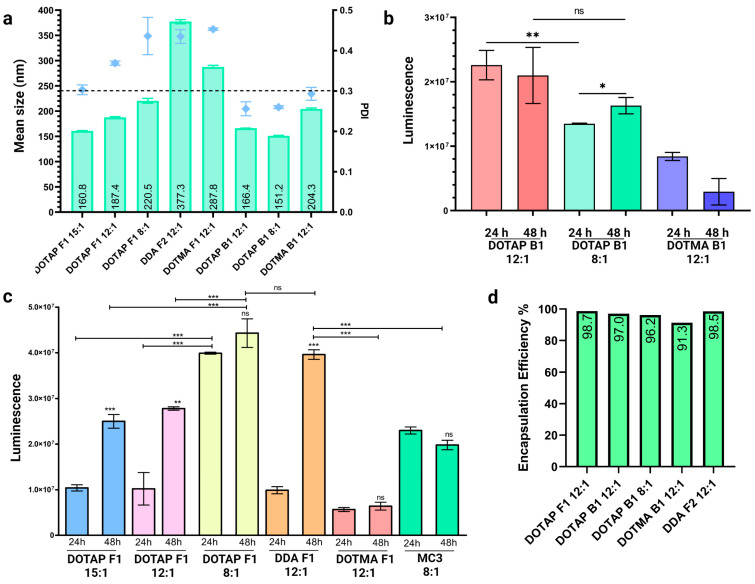
Formulation of saRNA-Int-cLNPs using a modified solvent injection method. (**a**) saRNA-Int-cLNPs were formulated using a modified solvent injection method. This method of incorporating saRNAs inside the cLNP generated particles that were generally larger, ranging from 151.2–377.3 nm, with sub-optimal PDIs. (**b**,**c**) Transfection of HEK293 cells with saRNA-Luc2-cLNP formulations comprising DOTAP, DOTMA, or DDA all achieved delivery of saRNA-Luc2. At 48 h post-transfection, luminescence signals increased in DOTAP B1, DOTAP F1, and DDA F1 suggesting amplification of the transcript was achieved. Once again, formulation composition, as well as N:P ratios were vitally important for optimal delivery and expression kinetics. saRNA-cLNPs formulated using the solvent injection method outperformed a standard MC3-based saRNA-iLNP formulation prepared using microfluidics. (**d**) As with exteriorly formulated saRNAs, interiorly formulated saRNAs showed high (>90%) encapsulation efficiency. Results are depicted as the mean, with standard error of the mean. Significant differences were calculated using a Student’s *t*-test. ns = not significant, * *p* < 0.05, ** *p* < 0.01, *** *p* < 0.001. Created using GraphPad prism 4 and 5 and BioRender.com.

## Data Availability

Data is contained within the article or Appendix A.

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
