# Peer review of "Production, Characterization, and Assessment of Permanently Cationic and Ionizable Lipid Nanoparticles for Use in the Delivery of Self-Amplifying RNA Vaccines"

_pharmaceutics, 2023, doi:10.3390/pharmaceutics15041173_

Round 1
Reviewer 1 Report
1. the methodology part needs to be mentioned in detail
2. gel retardation assay is essential to study the effect of nanoparticles stability
3. it is highly recommended to do serum stability and RNAse stability
Reviewer 2 Report
In this work, the authors indicated that CCAU-saRNAs-Luc2 does indeed trigger the innate immune system in a dose-dependent manner, and compared the ability of different LNPs to deliver saRNA transcripts while successfully applying a modified solvent injection method to formulate saRNAs internally in cLNPs. The concept of this study is interesting and sound, with presenting a series of results for supporting the point of authors. But there are several issues before it is accepted.
1. The sequence label 'a b c d' in each figure needs to be enlarged a little.
2. In figure 2, the INF-β induced by saRNA-Luc2 at a dose of 500ng is significantly higher than Poly I:C, can you explain why?
3. The axes in figure2 (3c) need to be bolded.
4. The size of the scale bar corresponding to the figure in Figure4 should be clearly marked in the figure legend.
5. When comparing CaliVax nanoparticles and DOTMA F1 cLNPs in Figure 4a, it should be done with the same N:P ratio.
6. The images of DOTMA F1(12:1) in figure4a should be quantified to prove that eGFP expression increases with time in HEK293.
Round 2
Reviewer 1 Report
the authors have addressed all the comments